# CT Lymphography Using Lipiodol^®^ for Sentinel Lymph Node Biopsy in Early-Stage Oral Cancer

**DOI:** 10.3390/jcm11175129

**Published:** 2022-08-31

**Authors:** Rutger Mahieu, Dominique N. V. Donders, Jan Willem Dankbaar, Remco de Bree, Bart de Keizer

**Affiliations:** 1Department of Head and Neck Surgical Oncology, University Medical Center Utrecht, 3584 CX Utrecht, The Netherlands; 2Department of Radiology, University Medical Center Utrecht, 3584 CX Utrecht, The Netherlands; 3Department of Nuclear Medicine, University Medical Center Utrecht, 3584 CX Utrecht, The Netherlands

**Keywords:** mouth neoplasms, sentinel lymph node biopsy, lymphatic metastasis, lymphography, computed tomography, X-ray

## Abstract

This study evaluated sentinel lymph node (SLN) identification with CT lymphography (CTL) following peritumoral administration of Lipiodol^®^ relative to conventional ^99m^Tc-nanocolloid lymphoscintigraphy (including SPECT/CT) in 10 early-stage oral cancer patients undergoing SLN biopsy. Patients first underwent early dynamic and static scintigraphy after peritumoral administration of ^99m^Tc-nanocolloid. Subsequently, Lipiodol^®^ was administered at the same injection sites, followed by fluoroscopy and CT acquisition. Finally, late scintigraphy and SPECT/CT were conducted, enabling the fusion of late CTL and SPECT imaging. The next day, designated SLNs were harvested, radiographically examined for Lipiodol^®^ uptake and histopathologically assessed. Corresponding images of CT, ^99m^Tc-nanocolloid lymphoscintigraphy and SPECT/late CTL fusion were evaluated. ^99m^Tc-nanocolloid lymphoscintigraphy identified 21 SLNs, of which 7 were identified with CTL (33%). CTL identified no additional SLNs and failed to identify any SLNs in four patients (40%). Out of six histopathologically positive SLNs, two were identified by CTL (33%). Radiographic examination confirmed Lipiodol^®^ uptake in seven harvested SLNs (24%), of which five were depicted by CTL. CTL using Lipiodol^®^ reached a sensitivity of 50% and a negative predictive value (NPV) of 75% (median follow-up: 12.3 months). These results suggest that CTL using Lipiodol^®^ is not a reliable technique for SLN mapping in early-stage oral cancer.

## 1. Introduction

In early-stage oral squamous cell carcinoma (OSCC), sentinel lymph node biopsy (SLNB) has proven to be a reliable staging procedure for the clinically negative neck (cN0), with similar locoregional-free, disease-specific and overall survival rates as well as significantly lower functional morbidity compared to its alternative, elective neck dissection (END) [1,2,3].

Since the introduction of SLNB, sentinel lymph node (SLN) imaging has undergone several technologic refinements, including the introduction of single-photon emission computed tomography/computed tomography (SPECT/CT) [4]. Yet, both technical and logistical challenges remain for SLNB using conventional technetium-99m (^99m^Tc) labeled radiotracers, which instigate the ongoing investigation for alternative high-resolution SLN imaging techniques [5,6,7].

Among others, CT lymphography (CTL) using peritumorally administered water-soluble iodine-based contrast agents (e.g., iopamidol) has been put forward as a high-resolution alternative for conventional lymphoscintigraphy and has been studied in several tumor types, including early-stage OSCC [8,9,10,11,12].

However, owing to the high velocity of lymphatic transportation, limited retention and rapid washout of water-soluble iodine-based contrast agents [13], true SLNs may be overlooked, and contrast-enhanced higher-echelon nodes may be erroneously designated as SLNs. Sugiyama et al. found that 4/27 patients staged negative for nodal metastasis on the basis of CTL using iopamidol actually had occult cervical lymph node metastases. In these patients, overlooked SLNs were only marginally contrast-enhanced on CTL, resulting in a poor accuracy of the procedure (sensitivity: 56%, negative predictive value (NPV): 85%). Furthermore, re-evaluation of the CTL images showed that iopamidol was only briefly retained in SLNs, with Hounsfield units (HU) of the contrast-enhanced SLNs already decreasing after 10 min post-injection [14].

Lipiodol^®^, an oil-based iodinated contrast agent with a higher viscosity, may provide increased retention of SLNs as well as delayed tracer wash-out and, thus, might improve SLN identification on CTL in early-stage OSCC patients [13,15]. Kim et al. already demonstrated the feasibility of CTL using Lipiodol^®^ for SLN mapping in patients with early gastric cancer [13]. In their study, CTL successfully visualized contrast-enhanced SLNs at 1 h following the peritumoral submucosal injection of Lipiodol^®^ in all patients.

Furthermore, Lipiodol^®^ has been combined with indocyanine green (ICG) as a single emulsion, which could overcome the limitations of injecting two individual tracers consecutively (i.e., dual-tracer methods), potentially enabling the reliable intra-operative localization of pre-operative depicted SLNs [7,16].

Therefore, this study evaluated SLN identification with CTL following the peritumoral administration of Lipiodol^®^ compared to SLN identification using conventional lymphoscintigraphy, including SPECT/CT with ^99m^Tc-nanocolloid in early-stage OSCC patients. 

## 2. Materials and Methods

### 2.1. Patients

This study was approved by the Ethics Committee of the University Medical Center Utrecht (no. 20/079) and was registered in the Netherlands Trial Register (accessible via https://www.trialregister.nl/, registration number NL9005). 

Between November 2020 and June 2021, a total of 10 patients with newly diagnosed early-stage OSCC (cT1-2N0; TNM Staging AJCC UICC 8th Edition [17,18]), scheduled for SLNB, were prospectively included in this study.

In all patients, cN0 status was confirmed by palpation and ultrasound of the neck. In those with suspect lymph nodes on ultrasound, ultrasound-guided fine-needle aspiration cytology was performed. In most patients (80%), magnetic resonance imaging of the head and neck was acquired as part of clinical staging.

Patients with prior head and neck malignancies in the past 5 years were excluded from this study. In addition, patients with a history of gross injury to the neck that precluded the reasonable surgical dissection of SLNs for this trial, neck dissection or radiotherapy to the neck were excluded from this study as well. Finally, patients who previously had an allergic reaction after the administration of an iodine-based contrast agent or those with manifest hyperthyroidism could not participate in this study.

### 2.2. Study Design

Figure 1 provides an overview of the study’s procedures.

The day before surgery, included patients first underwent conventional early scintigraphy (i.e., dynamic and early static) following the peritumoral administration of 120 megabecquerel (MBq) ^99m^Tc-nanocolloid in a volume of ~0.5 mL by an experienced radiologist and nuclear medicine physician (B.d.K.).

Then, a total volume of 0.5 mL Lipiodol^®^ was administered peritumorally at the same injection site by the same physician (B.d.K.) in the presence of a second observer (R.M.; D.N.V.D.). In order to record the early lymphatic drainage of Lipiodol^®^, peritumoral administration of Lipiodol^®^ was immediately followed by a dynamic X-ray in odontoid view (projections, anterior–posterior and anterior–oblique; 75 kV, 220 mA, small focus, filter: 0.1 mm Cu + 1 mm Al) until 20 min post-injection (Figure 2). Subsequently, CTL images were acquired using IQon Spectral CT (Philips Healthcare, the Netherlands) (Figure 3). CT acquisition was conducted using the following settings: 120 kV; 62 mAs; rotation time: 0.27 s; slice thickness: 0.9 mm.

After CTL, late scintigraphy and SPECT acquisition were initiated 2 h post-injection of ^99m^Tc-nanocolloid and were followed by low-dose CT acquisition as part of SPECT/CT, according to the EANM guidelines [19]. Since the low-dose CT, as part of SPECT/CT, was conducted after the injection of Lipiodol^®^ (120–125 min post-injection), SPECT/CT acquisition inherently allowed for the fusion of late CTL and SPECT imaging (Figure 4).

SLNs identified by ^99m^Tc-nanocolloid lymphoscintigraphy (including SPECT/CT) were marked on the overlying skin. Any additional SLNs identified in vicinity of the primary tumor by CTL were marked for surgical extirpation as well on the basis of their anatomical location. 

The next day, the marked SLNs were harvested under conventional portable gamma-probe guidance. The location of the extirpated SLNs, including their counts per second, as measured with the portable gamma-probe, were registered. In those with tumors involving the floor-of-mouth, super-selective neck dissections of cervical lymph node level I were performed [20]. Any SLNs in the super-selective neck dissection specimen were extirpated ex vivo under gamma-probe guidance. 

All extirpated SLNs were subjected to low-dose X-ray using a mammography system (Hologic 3Dimension^TM^; target-filter rhodium, 30 kV, 10 mAs, small focus) to assess whether those SLNs contained Lipiodol^®^ (Figure 5). 

Finally, harvested SLNs were sent for histopathological examination using step-serial-sectioning (section thickness 150 μm) with hematoxylin-eosin and pan-cytokeratin antibody (AE 1/3) staining [21,22]. Patients with histopathologically negative SLNs were assigned to a wait-and-scan approach. In those with at least one histopathologically positive SLN, complementary treatment of the affected and adjacent nodal basins was employed (i.e., neck dissection or (chemo)radiotherapy). Complementary neck dissection specimens were histopathologically assessed for additional (non-SLN) nodal metastases. The follow-up of patients was in compliance with standard oncological care.

### 2.3. Preparation of Lipiodol^®^ for Peritumoral Injection

Lipiodol^®^ was obtained from Guerbet, Roissy, France. Before peritumoral administration, Lipiodol^®^ was pre-heated at 37 °C using a contrast media warming cabinet, having a viscosity of approximately 25 mPa·s [15]. A total volume of 0.5 mL Lipiodol^®^ was peritumorally injected using a 1 mL compatible syringe.

### 2.4. Scintigraphy and SPECT/CT

Planar scintigraphic images were acquired in dynamic mode (128 × 128 matrix; 60 frames of 30 s) in anterior–posterior projection, followed by a static mode (256 × 256 matrix; 4 m) in anterior–posterior and lateral projections (30 m and 2 h post-injection). Dynamic and static scintigraphic imaging was supplemented with 30 s flood field images. SPECT/CT was acquired on a 128 × 128 matrix (pixel spacing, 4.8 × 4.8 mm), with 128 angles, 20 s per projection, over a non-circular 360° orbit (CT: 110 kV, 40 mAs eff., slice thickness 1.2 mm). Both scintigraphic and SPECT/CT images were acquired using a Siemens Symbia T16 SPECT/CT scanner with “low- and medium energy” (LME) collimators to limit septal penetration and reduce shine-through [23]. SPECT/CT images were reconstructed using clinical reconstruction software (Siemens Flash3D, Siemens Healthineers, Erlangen, Germany), with attenuation and scatter correction (6 iterations, 8 subsets, and a 5mm Gaussian filter).

### 2.5. Evaluation and Analyses

For each patient, corresponding images of CTL and conventional ^99m^Tc-nanocolloid lymphoscintigraphy (including SPECT/CT), as well as fusion images of late CTL and SPECT, were evaluated for the similarity of depicted draining lymph node basins, and the number and location of identified SLNs. Furthermore, CTL and ^99m^Tc-nanocolloid lymphoscintigraphy were compared with respect to the visualization of lymphatic vessels transporting the radiotracer. SLNs identified with either CTL or ^99m^Tc-nanocolloid lymphoscintigraphy, as well as outcomes of low-dose X-ray of harvested SLNs, were related to the outcomes from histopathological assessment of extirpated SLNs and any complementary neck dissection specimens. All images were simultaneously reviewed by a radiologist/nuclear medicine physician (B.K.) and a second observer (R.M.).

All data were analyzed with IBM SPSS Statistics Version 28.0 (IBM Corp., Armonk, NY, USA). For categorical variables, the number of cases and percentages are presented. Continuous parametric variables are presented as mean (±SD); non-parametric variables are presented as median. With histopathological examination of extirpated SLNs as well as any complementary neck dissection specimens and follow-up as reference standard, both sensitivity and NPV were calculated for CTL. Statistical tests were considered unfeasible, due to the small number of patients included in this study.

## 3. Results

Patient and tumor characteristics of included patients are listed in Table 1. The most frequently involved tumor subsite was the oral tongue (40%). Tumors were clinically classified as T2 in the majority of patients (80%). A total of 32 SLNs were harvested (a median of 3 per patient), of which 6 contained metastases. In this population, 40% of patients had occult lymph node metastasis as detected by SLNB. Of those with histopathologically positive SLNs, three underwent complementary neck dissection (75%), while the remaining patients underwent complementary radiotherapy. No additional nodal metastases were found by the histopathological examination of complementary neck dissection specimens. Adverse reactions following the peritumoral injection of Lipiodol^®^ did not occur. The median follow-up time was 12.3 months (ranging between 5.6 and 15.6 months). 

### 3.1. Sentinel Lymph Node Identification

The outcomes of CTL and ^99m^Tc-nanocolloid lymphoscintigraphy for all included patients are presented in Table 2.

^99m^Tc-nanocolloid lymphoscintigraphy identified a total of 21 SLNs, of which 7 were identified by CTL as well (33%). No additional SLNs were identified by CTL. In four patients, CTL failed to identify any SLN (40%). Out of the six histopathologically positive SLNs, two were identified by CTL (33%).

Low dose X-ray confirmed the uptake of Lipiodol^®^ in seven harvested SLNs (24%), in a total of five patients, of which five SLNs were also depicted by CTL. Due to logistical issues, the extirpated SLNs of patient four were not assessed for the presence of Lipiodol^®^ using low-dose X-ray. 

Lymphatic vessels draining ^99m^Tc-nanocolloid towards SLNs were visualized in four patients by early dynamic and static scintigraphy (patients 2, 3, 5 and 6; 40%). In regard to CTL, lymphatic vessels draining Lipiodol^®^ were identified in three patients (patients 3, 6 and 9; 30%).

### 3.2. Follow-Up

During follow-up, one patient developed nodal recurrence 11 months following SLNB (patient 1). Since the nodal recurrence occurred in the nodal basin that was initially staged positive by SLNB and subsequently treated with complementary radiotherapy, it was considered to be a consequence of inadequate complementary treatment rather than SLNB failure. All other included patients remained free of disease. With a median follow-up of 12.3 months, CTL using Lipiodol^®^ reached a sensitivity of 50% and a NPV of 75% in this population.

Follow-up cross-sectional imaging was performed in two patients, according to the standard of oncological care, on account of a suspected recurrence (patient 1) or for further oncological staging after positive SLNB (patient 6) at 331 days and 13 days following SLNB, respectively. In both patients, remains of Lipiodol^®^ were still seen surrounding the initial primary tumor site on FDG-PET/CT (Figure 6). However, no further lymphatic drainage of Lipiodol^®^ nor depositions of Lipiodol^®^ in lymph nodes were observed in either patient on follow-up imaging.

## 4. Discussion

In this within-patient comparison study aimed at evaluating SLN mapping using CTL with Lipiodol^®^ in early-stage OSCC, a disappointing SLN identification rate of only 33% was found for CTL. Contrast-enhanced lymphatic vessels were identified in three patients by CTL (30%), while conventional early dynamic and static scintigraphy managed to visualize lymphatic vessels draining ^99m^Tc-nanocolloid in four patients (40%). Moreover, in 40% of patients, neither lymphatic drainage of Lipiodol^®^ nor contrast-enhanced lymph nodes were observed at all on CTL. In this population, CTL using Lipiodol^®^ would have falsely staged two patients as negative for nodal metastatic disease (patients 1 and 9), corresponding with a sensitivity of 50% and an NPV of 75%.

Hence, our results contradict those of Kim et al. [13], who identified at least one contrast-enhanced lymph node in all 10 early gastric adenocarcinoma patients by CTL one hour following the peritumoral injection of 1 mL Lipiodol^®^. Furthermore, in their study, all metastatic lymph nodes (*n* = 3) retained Lipiodol^®^ as confirmed by histopathological examination of the sentinel nodal basin including immunohistochemistry and Oil-Red-O staining [13]. Given that elective lymphadenectomy was performed in all patients, no step-serial-sectioning of ex vivo-dissected (S)LNs was performed (sectioning thickness 2 mm), non-sentinel basins were subjected to routine histopathological examination and no follow-up results were reported, the accuracy of CTL using Lipiodol^®^ as described by Kim et al. may actually be lower. Especially when considering that micro-metastases and isolated tumor cells are easily missed by routine histopathological examination of lymphadenectomy specimens or by sectioning SLNs at larger intervals than 150 μm [24,25,26,27]. Still, this does not entirely clarify the discrepancy in the rate of identified contrast-enhanced lymph nodes by CTL using Lipiodol^®^ between both studies. Although larger volumes of Lipiodol^®^ were administered by Kim et al., the effect of larger volumes (>0.5 mL) on SLN identification is dubious and has not been recommended by EANM guidelines, as it may lead to a collapse of lymphatics and an increase in patient discomfort [19,28,29]. Presumably, differences in characteristics of the lymphatic system between tumor sites contribute to these contrasting results. 

Despite the fact that peritumorally administered Lipiodol^®^ exhibited poor lymphatic drainage in our population, its capabilities in terms of long-term retention and delayed tracer wash-out were demonstrated by both late CTL and low-dose X-ray of harvested SLNs. In fact, follow-up cross-sectional imaging in two patients showed retention of Lipiodol^®^ at 13 days and 11 months post-injection. However, it should be pointed out that these remains of Lipiodol^®^ were only seen surrounding the initial primary tumor site and that no further lymphatic drainage nor depositions of Lipiodol^®^ in lymph nodes were observed. These findings on the long-term retention of Lipiodol^®^ are in line with Kim et al., which revealed that extirpated lymph nodes still contained Lipiodol^®^ one day following CTL [13].

Overall, our results suggest that CTL using Lipiodol^®^ is inferior to CTL using water-soluble iodine-based contrast agents. In previous studies investigating CTL using a water-soluble iodine-based contrast agent (i.e., iopamidol), at least one SLN could be identified by CTL in 89–96% of patients [7,8,9,10,11,12,14]. In addition, two series reported contrast-enhanced lymphatic vessel visualization in 90% of their patients [9,11]. Their approach for SLNB using CTL led to a reported sensitivity ranging from 56 to 80% and a NPV ranging from 82 to 96% [7,9,10,12,14]. Nonetheless, it should be mentioned that in some of these studies, SLNB was immediately followed by END, irrespective of the histopathological status of harvested SLNs [9,10]. As aforementioned, micrometastases are easily missed by routine histopathological examination of neck dissection specimens; therefore, the reported false-negative rate of SLN mapping by CTL in these studies may be underestimated [24,25,26]. In those studies with more reliable reference standards (i.e., observation of the untreated neck during follow-up after negative SLNB), the reported diagnostic accuracy of SLNB on the basis of CTL using iopamidol was significantly worse (sensitivity 56–63%, NPV 85–86%) [12,14]. For now, both our data as well as results from previous studies suggest that the strategy of SLNB with CTL, using either Lipiodol^®^ or water-soluble iodine-based contrast agents (e.g., iopamidol), is inferior to SLNB with conventional ^99m^Tc-labeled radiotracers [30].

Even so, modifications of oil-based iodinated contrast media could increase the degree of lymphatic drainage while still allowing for long-term retention in lymph nodes, potentially improving SLN detection by CTL [16,31]. Alternatively, other radiocontrast agents (e.g., Nanotrast-CF800 entrapped gold nanoparticles) have shown their potential for SLN mapping using CTL in animal studies [32,33,34]. These modified radiocontrast agents may enhance the utility of CTL for SLN mapping but have yet to be investigated in human studies.

Several limitations of this study have to be acknowledged. First of all, as previously mentioned, in this study a smaller volume of iodinated contrast agent was administered compared to other studies [9,10,11,12,13,14]. This discrepancy, as well as other dissimilarities (i.e., tumor site and reference standard), contribute to the fact that our results cannot easily be compared with those achieved by Kim et al. [13]. However, in our population, the vast majority of the administered Lipiodol^®^ remained in the injection site as depicted with CTL, late CTL and follow-up imaging. Accordingly, the marginal lymphatic drainage of Lipiodol^®^ itself, rather than the injected volume, appears to impede adequate SLN visualization. In addition, by using similar volumes for CTL and conventional ^99m^Tc-nanocolloid lymphoscintigraphy (i.e., ~0.5 mL), a more reliable comparison in regard to lymphatic drainage and SLN identification between both techniques could be actualized.

Secondly, the histopathological examination of extirpated SLNs did not include oil-staining to detect microscopic traces of Lipiodol^®^. Even though this did not impact the accuracy of CTL, it might have provided more insight into the (microscopic) lymphatic distribution of Lipiodol^®^. 

Finally, due to the small population size of this study and its limited follow-up data, the results of this study should be interpreted with caution. Especially since longer follow-up might reveal missed occult metastases as they become clinically manifest (false-negative outcome). 

## 5. Conclusions

In conclusion, the results of this study suggest that CTL following the peritumoral administration of Lipiodol^®^ is not a reliable technique for SLN mapping in early-stage OSCC, on account of the low SLN identification rate compared to the current standard for SLN mapping: lymphoscintigraphy including SPECT/CT using ^99m^Tc-labeled radiotracers. Although modified radiocontrast agents may enhance the utility of CTL for SLN mapping, they have yet to be further investigated in human studies.

## Figures and Tables

**Figure 1 jcm-11-05129-f001:**
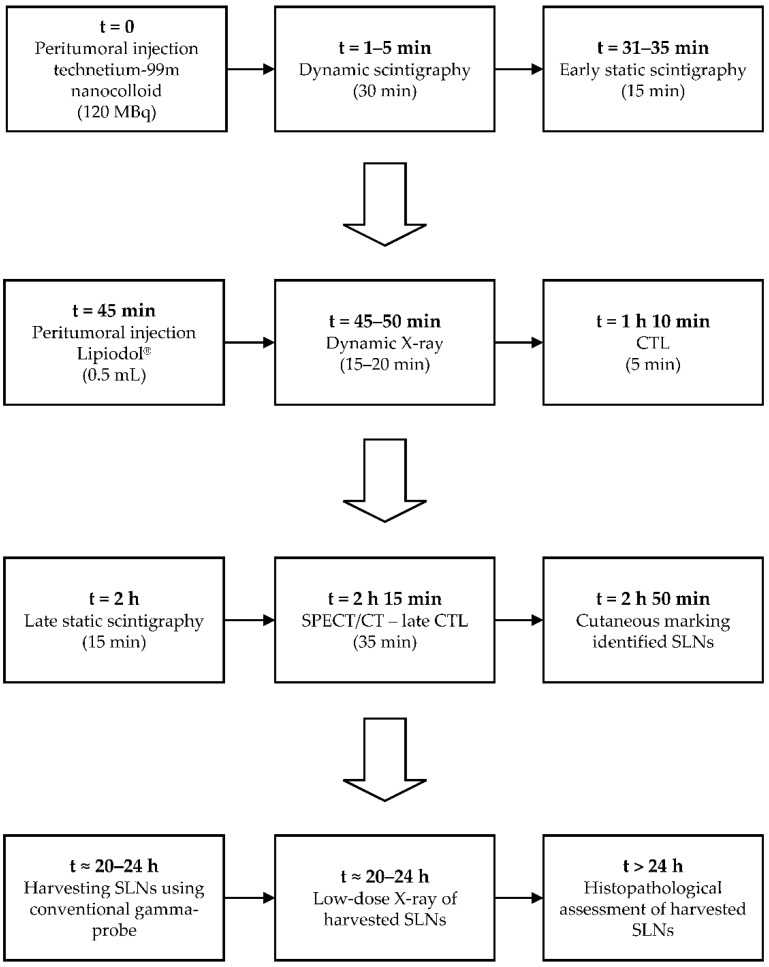
Time schedule of study events. MBq, megabecquerel; mL, milliliter; CTL, computed tomography lymphography; SPECT/CT, single-photon emission computed tomography/computed tomography; SLNs, sentinel lymph nodes.

**Figure 2 jcm-11-05129-f002:**
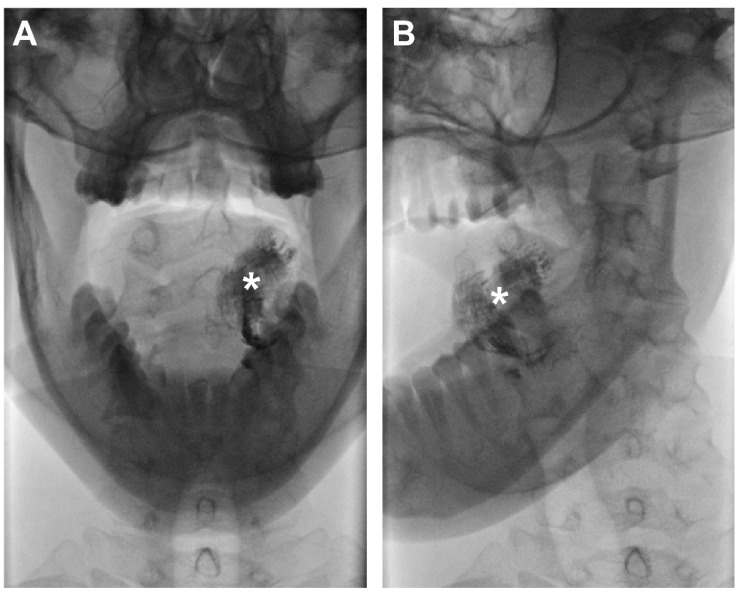
Contrast-inverted dynamic X-ray images acquired 10 min following peritumoral injection of Lipiodol^®^ in a patient with cT1N0 OSCC on the left side of the oral tongue (patient 6). No lymphatic drainage of Lipiodol^®^ was observed with dynamic X-ray. (**A**) Anterior–posterior odontoid projection; (**B**) anterior–oblique odontoid projection; (*) injection site.

**Figure 3 jcm-11-05129-f003:**
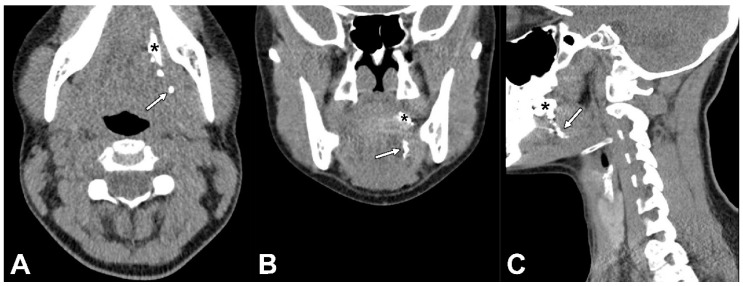
CT lymphographic images acquired 25 min post-injection of Lipiodol^®^ in the same patient (patient 6). (**A**) Axial plane, (**B**) coronal plane, and (**C**) sagittal plane; (*) injection site. Lymphatic drainage was observed (white arrows), yet no SLNs were identified on CTL in this patient.

**Figure 4 jcm-11-05129-f004:**
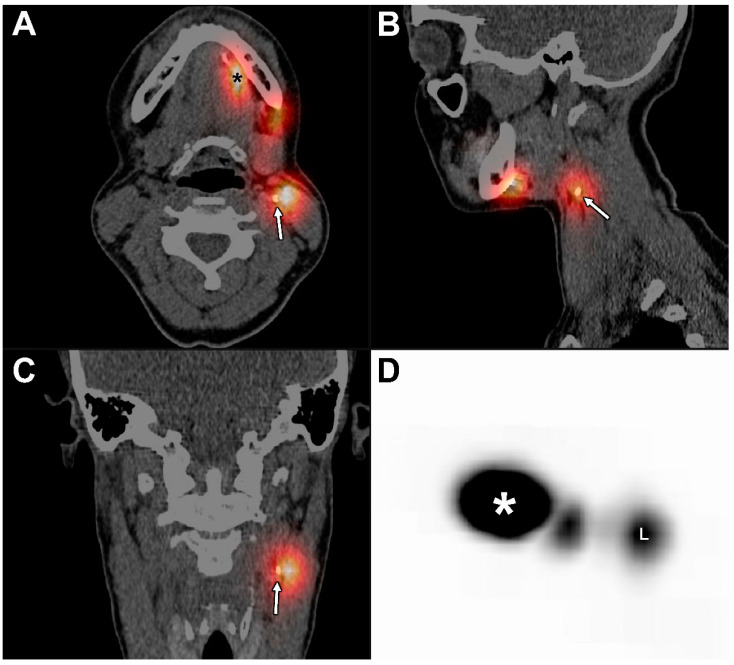
Fusion of SPECT and late CTL at 120 min post-injection of Lipiodol^®^ in the same patient (patient 6). (**A**) Axial plane, (**B**) sagittal plane, (**C**) coronal plane, and (**D**) maximum intensity projection (MIP); (*) injection site. The presence of Lipiodol^®^ was seen in a SLN located in cervical lymph node level IIa on the left side (white arrows), as identified with SPECT, corresponding with the hotspot (L) on MIP. Conventional ^99m^Tc-nanocolloid lymphoscintigraphy (including SPECT/CT) identified another SLN in cervical lymph node level Ib on the left side, in which no uptake of Lipiodol^®^ could be seen on late CTL.

**Figure 5 jcm-11-05129-f005:**
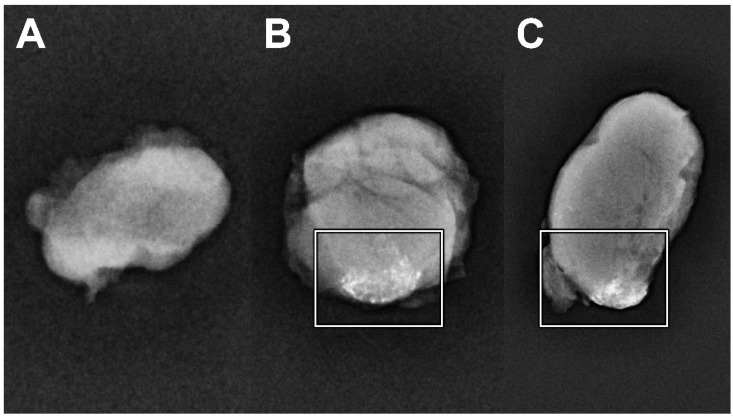
Low-dose X-ray of extirpated SLNs from patient 6. (**A**) SLN level Ib left (750 counts per second), which proved histopathologically positive for metastasis, showed no traces of Lipiodol^®^. (**B**,**C**) SLN level Ib left (5387 counts per second; histopathologically negative) and SLN level IIa left (9194 counts per second; histopathologically positive) with the uptake of Lipiodol^®^ as confirmed by low-dose X-ray (white squares).

**Figure 6 jcm-11-05129-f006:**
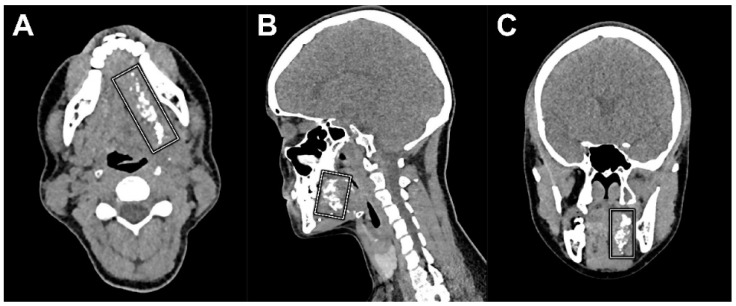
CT images as part of the FDG-PET/CT acquired 13 days following peritumoral administration of Lipiodol^®^ (patient 6) clearly depicts remains of Lipiodol^®^ located around the initial primary tumor site (white squares). (**A**) Axial plane, (**B**) coronal plane, and (**C**) sagittal plane.

**Table 1 jcm-11-05129-t001:** Patient- and tumor characteristics.

Characteristics	*n* = 10
Gender, *n* (%)	
Female	6 (60%)
Male	4 (40%)
Median age (y) (range)	67.0 (20–81)
History of head and neck cancer, *n* (%)	0 (0%)
Tumor location, *n* (%)	
Tongue	4 (40%)
Floor-of-mouth	2 (20%)
Buccal mucosa	3 (30%)
Vestibule of mouth	1 (10%)
Side primary tumor, *n* (%)	
Left	3 (30%)
Right	6 (60%)
Midline	1 (10%)
Clinical T-stage, *n* (%) *	
T1	2 (20%)
T2	8 (80%)
Pathology primary tumor	
Mean diameter (mm) (SD)	14.8 (±8.7)
Mean depth-of-invasion (mm) (SD)	4.0 (±2.6)
Median harvested SLNs (range)	3 (2-5)
Histopathological status SLNs, *n* (%)	
Negative	26 (81%)
Positive	6 (19%)
Pathological N-stage after SLNB, *n* (%) *	
pN0(sn)	6 (60%)
pN1(sn)	2 (20%)
pN2b(sn)	2 (20%)
Complementary neck treatment, *n* (%)	
Complementary ND	3 (30%)
Complementary RT	1 (10%)
Follow-up in months (range)	12.3 (6–16)

*n*, number; y, years; mm, millimeters; SD, standard deviation; SLNs, sentinel lymph nodes; SLNB, sentinel lymph node biopsy; ND, neck dissection; RT, radiotherapy. * According to the AJCC TNM classification, 8th edition.

**Table 2 jcm-11-05129-t002:** Comparison of sentinel lymph node distribution between CT lymphography and ^99m^Tc-nanocolloid lymphoscintigraphy.

Nº	Primary Tumor	Identified SLNs SG and SPECT/CT		Identified SLNs CTL		Harvested SLNs (cps)		PA	Lipiodol^®^ in Harvested SLN	Complementary Treatment	pTNM *
1	Lower gum (right)	Ib	Right	None		Ib	Right (360)	+	No	RT I–V right	pT4aN2b(sn)
		IIa	Right			Ib	Right (259)	+	No		
						IIa	Right (198)	-	No		
2	Tongue (right)	III	Right	III	Right	III	Right (2347)	-	Yes	N.A.	pT2N0(sn)
		III	Right	III	Right	III	Right (2004)	-	Yes		
		III	Left			III	Left (129)	-	No		
3	FOM (midline)	Ib	Right	III	Left	Ib	Right (171)	-	No	N.A.	pT1N0(sn)
		III	Right			III	Right (429)	-	Yes		
		III	Left			III	Left (411)	-	No		
						III	Left (328)	-	No		
4	FOM (midline)	Ib	Right	Ib	Right	Ib	Right (701)	+	N.S.	SND I–III right	pT1N1
		IIa	Right			IIa	Right (174)	-	N.S.		
		III	Left			III	Left (276)	-	N.S.		
5	Tongue (left)	III	Left	None		III	Left (447)	-	No	N.A.	pT1N0(sn)
		IV	Left			III	Left (441)	-	No		
						IV	Left (927)	-	No		
6	Tongue (left)	Ib	Left	IIa	Left	Ib	Left (750)	+	Yes	MRND I–V left	pT1N2b
		IIa	Left			Ib	Left (5387)	-	No		
						IIa	Left (9194)	+	Yes		
7	Buccal mucosa (right)	Ib	Right	Ib	Right	Ib	Right (97)	-	No	N.A.	pT1N0(sn)
		II	Right			Ib	Right (13)	-	No		
						Ib	Right (411)	-	No		
						Ib	Right (432)	-	No		
						II	Right (310)	-	Yes		
8	Buccal mucosa (left)	IIa	Left	None		IIa	Left (184)	-	No	N.A.	pT2N0(sn)
						IIa	Left (228)	-	No		
						II	Left (73)	-	No		
9	Tongue (right)	II	Right	II	Right	IIb	Right (963)	-	Yes	MRND I–V right	pT2N1
		III	Right			III	Right (363)	+	No		
						V	Right (648)	-	No		
10	Buccal mucosa (right)	III	Right	None		III	Right (23)	-	No	N.A.	pT1N0(sn)
						III	Right (158)	-	No		

Nº; patient number; SLN, sentinel lymph node; SG, scintigraphy; SPECT/CT, single-photon emission computed tomography/computed tomography; CTL, computed tomography lymphography; cps, counts per second as measured with conventional gamma-probe; PA, pathological assessment; +, histopathologically positive for metastasis; -, histopathologically negative for metastasis; RT, radiotherapy; N.A., not applicable; FOM, floor-of-mouth; MRND, modified radical neck dissection; SND, selective neck dissection. * Final pathological TNM-stage following surgery, SLNB and any complementary treatment (AJCC TNM classification, 8th edition).

## Data Availability

The data presented in this study are available upon request from the corresponding author.

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
