# Peer review of "CT Lymphography Using Lipiodol^®^ for Sentinel Lymph Node Biopsy in Early-Stage Oral Cancer"

_jcm, 2022, doi:10.3390/jcm11175129_

Round 1

Reviewer 1 Report

The topic of the manuscript is to evaluate sentinel lymph node (SLN) identification by CT lymphography following peritumoral administration of Lipiodol® relative to conventional 99mTc-nanocolloid lymphoscintigraphy (including SPECT/CT) in 10 early-stage oral cancer patients undergoing SLN biopsy.

The title and the abstract of the article are informative. The Introduction briefly presents the issue of sentinel lymph node biopsy, as well as Lipiodol®. The section "Material and Methods" precisely describes the chosen study design. The section "Results" is well prepared. The Discussion is interestingly written, including the recent references and the study limitations. The Conclusions seem to be the "take-home" messages.

Some following points must be clarified/corrected for the further processing of this article.

Merits-related comments:

1.       Please complete keywords with the proper MeSH terms.

2.       The weakest point of the study is the very small sample size. How can it be justified?

3.       In the Introduction or the Discussion, the Authors could mention recent innovative reports on the non-invasive early detection of oral squamous cell carcinomas using e.g. salivary metabolites (proposed references: 10.3390/metabo12040294, 10.3390/metabo11090587).

Technical comments:

1. References should be described as follows:
1. Author 1, A.B.; Author 2, C.D. Title of the article. 
Abbreviated Journal Name YearVolume, page range.

Author Response

Reviewer #1

The title and the abstract of the article are informative. The Introduction briefly presents the issue of sentinel lymph node biopsy, as well as Lipiodol®. The section "Material and Methods" precisely describes the chosen study design. The section "Results" is well prepared. The Discussion is interestingly written, including the recent references and the study limitations. The Conclusions seem to be the "take-home" messages

We would like to thank the reviewer for reading and reviewing our manuscript and for his/her generous comments.

  1. Please complete keywords with the proper MeSH terms.

We would like to thank the reviewer for his/her acuity. The keywords have been adjusted according to the proper MeSH terms as registered in the NCBI MeSH database.

  1. The weakest point of the study is the very small sample size. How can it be justified?

We fully agree with the reviewer that the small sample of this study represents one of its major weaknesses. This study was designed as a prospective cohort study including a total of 94 patients. The sample size of the full study was based on a non-inferiority design, the sensitivity of SLNB in floor-of-mouth cancer patients (0.67) and the sensitivity of SLNB in OSCC-patients excluding floor-of-mouth cancer patients (0.86) as reported by den Toom et al. [1]. However, before initiating the full study, this pilot was conducted to assess the utility of CT lymphography in 10 early-stage oral squamous cell carcinoma patients. The proposed manuscript presents the results of the pilot study. Due to the disappointing results of the pilot study in regard of CT lymphography, considering its very poor SLN detection rate compared to conventional 99mTc-nanocolloid lymphoscintigraphy, commencing with the full study was not justifiable.

  1. In the Introduction or the Discussion, the Authors could mention recent innovative reports on the non-invasive early detection of oral squamous cell carcinomas using e.g. salivary metabolites (proposed references: 10.3390/metabo12040294, 10.3390/metabo11090587).

Although intrigued by the utilization of salivary metabolites to diagnose and monitor staging in patients with OSCC, the authors are not confident that discussing its application fits this manuscript. Especially since the aim of SLNB is to detect occult lymph node metastases, whereas no differences were seen in salivary metabolites depending on the presence of lymph node metastases in OSCC patients [2].

  1. References should be described as follows: 1. Author 1, A.B.; Author 2, C.D. Title of the article. Abbreviated Journal Name Year, Volume, page range.

Again we would like to thank the reviewer for his punctiliousness. The references have been adjusted accordingly.

References

  1. den Toom, I.J.; Boeve, K.; Lobeek, D.; et al. Elective Neck Dissection or Sentinel Lymph Node Biopsy in Early Stage Oral Cavity Cancer Patients: The Dutch Experience. Cancers (Basel). 2020 Jul 3;12(7):1783. doi: 10.3390/cancers12071783.
  2. Lohavanichbutr, P.; Zhang, Y.; Wang, P.; et al. Salivary Metabolite Profiling Distinguishes Patients with Oral Cavity Squamous Cell Carcinoma from Normal Controls. PLoS ONE 2018, 13, e0204249.

Reviewer 2 Report

The manuscript made by Mahieu R et al is interesting and well done; they studied the efficacy of Lipiodol in lymph node of oral cancer.

I have a few questions that authors need to resolve.

The images reviewed by B. K and R. M. were made at the same time?
Would it be possible to establish a kappa study between observers (blind study)

Only were evaluated women?
I did not understand table 1, into methodology, were evaluated 10 patients, but in this table only were studied 6 patients, please review, and explain.
Would be possible to include histopathologic studies.
For example
Histopathological studies
oral carcinoma, well differentiated, or mild or poor differentiated, or spindle cell carcinoma, etc.

On the title authors have written studies in early-stage cancer, but on table 2 they have early, and advanced stage please explain better.

Discussion
Line 250 to 254, "Hence..." Studies made by Kim et al were gastric cancer; does there exist a possibility that oral cancer and gastric cancer had at same distribution of lipiodol? What type of cancers were studied in the article by Kim? Please explain better this paragraph.

Author Response

Reviewer #2

The manuscript made by Mahieu R et al is interesting and well done; they studied the efficacy of Lipiodol in lymph node of oral cancer.

We would like to express our gratitude to the reviewer for reading and reviewing our manuscript and for his/her generous comments.

  1. The images reviewed by B.K. and R.M. were made at the same time? Would it be possible to establish a kappa study between observers (blind study).

We would like to thank the reviewer for this insightful suggestion, as the authors fully agree with the reviewer that a blind study could have been highly interesting. Unfortunately, the images were reviewed by both observers at the same time, meanwhile discussing their outcome. Therefore, an interobserver kappa cannot be established. The manuscript has been adjusted to clarify that images were reviewed simultaneously by both observers. 

  1. Only were evaluated women? I did not understand table 1, into methodology, were evaluated 10 patients, but in this table only were studied 6 patients, please review, and explain.

We acknowledge the reviewers concerns. Since the heading of Table 1 mentions the number of included patients (n=10), the authors initially refrained from specifically mentioning the number of male patients included (n=4) as well. However, to avoid any further confusion, the number of male patients included has been added to Table 1.

  1. Would be possible to include histopathologic studies. For example oral carcinoma, well differentiated, or mild or poor differentiated, or spindle cell carcinoma, etc.

We would like to thank the reviewer for his/her suggestion. The authors agree that information on the grade of tumor cell differentiation can be meaningful, especially when reporting on patient’s prognosis. Nevertheless, since this study was mainly focused on the detection of SLNs using CT lymphography compared to conventional 99mTc-nanocolloid lymphoscintigraphy, it is the considered opinion of the authors that including histological grading is not of additional value for this particular diagnostic study and may even distract the reader from its relevant outcomes. In future studies, when reviewing recurrence-free survival and disease-specific survival, histological grading will certainly be included.

  1. On the title authors have written studies in early-stage cancer, but on table 2 they have early, and advanced stage please explain better.

Patients were included based on clinical TNM-staging (cT1-2N0). The pTNM as mentioned in Table 2 represents the final pathological TNM stage following surgery and any complementary treatment. In some cases patients appeared to have advanced staged disease as confirmed by histopathological assessment, while being clinically diagnosed as early-stage disease. For instance, patient 1 was clinically staged as cT2N0, however following surgery, including SLNB, histopathological examination revealed that the primary tumor showed signs of mandibular invasion (pT4a) and that two SLNs actually harbored metastasis (pN2b). We devoted a small passage to the Table Legends to clarify that the pTNM-stage as mentioned in Table 2 represents the final pathological TNM stage following surgery, SLNB and any complementary treatment.

  1. Line 250 to 254, "Hence..." Studies made by Kim et al were gastric cancer; does there exist a possibility that oral cancer and gastric cancer had at same distribution of lipiodol? What type of cancers were studied in the article by Kim? Please explain better this paragraph.

As the lymphatic system of the (epi)gastric region is vastly different from the cervical lymphatic system, the results of Kim et al. and our results cannot be easily compared, as already elucidated in our Discussion section. There is no information on differences in lymphatic distribution between types of cancers (e.g. adenocarcinoma, squamous cell carcinoma), although some types of cancer indeed have a higher tendency to spread (e.g. via the lymphatic system), compared to others. Therefore, information on the type of cancer studied in the article by Kim et al. (gastric adenocarcinoma) has been added in the corresponding paragraph.